# Effect of emotional regulation on performance of shooters during competition: An ecological momentary assessment study

**Zhou Lulu**[1,2☯], **Liu Huimin**[2], **Su Hua**[1☯]*

**1** School of Education, Hengshui University, Hengshui, China, **2** School of Psychology, Beijing Sport University, Beijing, China

☯ These authors contributed equally to this work.
* 83463689@qq.com

## Abstract

### Objective

The purpose of this study was to examine the effect of emotion regulation on shooting performance in shooting athletes.

### Method

Ecological momentary assessment was used to track and examine the dynamic relationship between pre-competition and in-competition emotions, emotion regulation strategy selection and use, and shooting performance in 57 shooting athletes.

### Results

Female athletes used more regulation strategies than male athletes and had lower mean shooting scores when using disengagement strategies and lower good ten-ring percentages when using engagement strategies compared to male athletes. Elite athletes had a higher percentage of ten-ring scores when using engagement strategies than did first-level athletes, but a lower percentage of ten-ring scores when using disengagement strategies. The use of emotion regulation strategies and situational demands were not strongly related. Athletes had low flexibility in emotion regulation and were better at using disengagement strategies, but disengagement strategies were not beneficial for shooting performance.

### Conclusions

First-level athletes have a higher contextual demand for emotion regulation and tend to use disengagement strategies more frequently to regulate emotions. Elite-level athletes have higher average ring values and ten-ring ratios when using engagement strategies under lower contextual demand and using disengagement strategies under higher contextual demand. Enhancing flexible emotional regulation training that improves situation-strategy fit may be beneficial for enhancing sports performance.

**Data availability statement:** The dataset for this study is available on Github, and the direct link to freely access the data set is https://github.com/ZhouLulu-cloud/Effect-of-Emotional-Regulation-on-Performance-of-Shooters-during-Competition.git.

**Funding:** Psychological Capacity Evaluation and Talent Selection Research of High-level Shooting and Archery Athletes" (2024JT07). The funders played a significant role in sourcing data for this paper by connecting us with collaborative units for receipt collection and program implementation.

**Competing interests:** The authors have declared that no competing interests exist.

# 1. Introduction

Shooting is a precise and static sport. It requires a high level of psychological stability. The performance of specialized skills such as stability and coordination in shooting sports depends to a certain extent on the conscious state of the shooter during the execution of the action [1]. Especially in shooting sports that require extremely high accuracy, the difference in skill level between shooters is minimal, resulting in great suspense and intense emotional changes caused by changes in performance [2,3]. Negative emotions also have a direct impact on shooting performance [3]. High-intensity emotional experiences can affect cognition, self-control, perception, action, coordination, and strategy, resulting in errors [3,4]. For example, a significant increase in anxiety level before a competition, continuous fluctuations during the competition, and severe impact on the performance of sports skills, even leading to action errors. Physical anxiety has a very strong effect on shooting performance [4]. Elite shooters believe that the level of anxiety will increase significantly before the competition and will continue to fluctuate during the competition, which is an important psychological factor affecting the Olympic shooting competition [5]. Athletes are under great psychological pressure, which causes athletes' psychological anxiety and stress, which has a negative impact on athletes' performance [6]. Therefore, the ability to regulate emotions at a higher level is a prerequisite for stable shot release and an important factor in the shooter's ultimate ability to compete [7,8].

## 1.1. The effect of emotion regulation strategies on sport performance

Emotion regulation is often considered to be an important component of an athlete's psychological abilities. The primary focus is on how individuals use regulation strategies and the effectiveness of these strategies to reflect the impact of emotion regulation on an individual's psychological activity and behavior. Cognitive emotion regulation strategies related to competition (such as planning for reflection, positive reappraisal, and acceptance) reflect the emotion regulation ability of active shooting athletes and promote shooting performance [9]. Emotion regulation plays a primary role in different time periods during the process of emotion generation, where individuals use regulation strategies to regulate their emotional state and intensity at different times [10]. Cognitive reappraisal (changing one's understanding of events and situations) and expressive suppression (attempting to suppress emotions) represent two specific types of regulation strategies. They are the most commonly used emotion regulation strategies in laboratory studies. Cognitive reappraisal has also been found to be a broadly applicable regulation strategy for all types of emotional situations [10]. In the field of sport, cognitive reappraisal has a better effect on regulating pre- and post-competition emotions, mainly by reducing the interference caused by pre-competition excitement and anxiety, and positively predicting competition performance. It also has a positive effect on post-traumatic stress and competition anxiety [11,12]. However, the regulatory effect of reappraisal during the competition process is inconsistent. It can effectively improve golfers' shooting rate and accuracy [13]. Meanwhile, some studies have found that reappraisal of physiological arousal can promote adaptive stress responses (cardiovascular responses) and increase a higher sense of resources and confidence [14]. During athletic movement performance, expressive suppression is more effective than cognitive reappraisal and distraction in promoting reaction time improvement and reducing movement errors within the transient activated period of emotion regulation (5-8s) [15]. Compared to emotional suppression, emotional physiological response suppression (biofeedback) can more effectively improve athletic performance [16]. Expressive suppression is the most commonly used strategy by athletes in intense competition with limited time [15], but the effective time is short, and the negative experience caused by high intensity may disrupt the movement structure. When experiencing low-intensity negative emotions, expressive

suppression may slow down the grip force in shooting, risking missed opportunities to fire, yet it paradoxically boosts shooting accuracy [17]. It can be seen that the effect of specific emotion regulation strategies is not stable. Athletes in competitive sports do not merely employ a specific strategy; instead, they select different emotion regulation strategies based on the changing context [9,15,16]. The prerequisite for effective emotion regulation strategies is the context that triggers emotional changes. Previous studies have focused on exploring regulation strategies or patterns that are applicable to sport situations, and compared to life situations, sport situations are constantly changing, and the applicability of specific regulation strategies is challenged.

## 1.2. Classification of emotion regulation strategies

Previous research has classified emotion regulation strategies into adaptive strategies (problem solving, reappraisal, mindfulness) and maladaptive strategies (expressive suppression, avoidance, rumination) based on their adaptability to emotions triggered by life situations [18]. This classification is more in line with the scope of research in health psychology, while the sport field focuses on investigating emotion regulation strategies that improve sport performance. The ability model suggests that emotional clarity, emotional tolerance, and emotion regulation flexibility are the potential for individuals to regulate emotions [10,18]. They are all important factors of emotional regulation ability, especially emotion regulation flexibility, which reflects the individual's ability to flexibly choose strategies in changing situations. Regulation strategies are divided into engagement and disengagement strategies based on the degree of successful emotion regulation and tolerance of aversive emotions [19]. Characteristics of engagement strategy is active participation in the pain or confronting the root cause of the emotion, including cognitive reappraisal, acceptance, and problem solving. Characteristics of the disengagement strategy include strong avoidance of painful emotions, thoughts, or situations, including expressive suppression and avoidance. Rumination is a form of cognitive processing where an individual repeatedly dwells on negative aspects of their experiences, thoughts, feelings, or problems without taking active steps to resolve them. This process is characterized by a passive disengagement from the situation, often accompanied by a tendency to avoid confronting or dealing with the difficulties head-on. Rumination (disengagement with a tendency to avoid difficulties) is also classified as a disengagement strategy [20]. Strategies were drawn from meta-analytical work that identified six distinct, commonly used strategies: three engagement strategies (cognitive reappraisal, acceptance, problem solving) and three disengagement strategies (rumination, expressive suppression, avoidance) [19,22]. Regulation strategies that successfully regulate emotions or tolerate aversive emotions are better suited for athletes to achieve their goal of enhancing sport performance than regulation strategies that are more adaptable to individual psychological health development [15,18,19].

## 1.3. Flexible use of emotion regulation strategies

Emotion regulation plays an important role in the performance of athletic skills. Research in sport psychology has attempted to identify patterns similar to those in the mental health field, namely, which regulation patterns and strategies can help athletes effectively regulate negative emotions in the competitive arena and maintain normal performance levels. However, in rapidly changing competitive sports situations (e.g., Changes in performance, such as leading or lagging behind, can trigger intense emotional experiences for athletes.), the effects of specific emotion regulation strategies are not stable, and athletes need to flexibly choose and use different strategies based on changes in the situation to achieve good results. Psychological research on emotion regulation has gradually shifted from investigating the adaptability

of specific emotion regulation strategies to investigating the flexibility of regulation strategies that balance context and strategy [21]. The ability to dynamically adjust regulation strategies based on changing environmental demands is referred to as emotion regulation flexibility [22]. Therefore, assessing environmental demands is a prerequisite for choosing or switching regulation strategies. An individual's environmental demands are influenced by emotional intensity and perceived controllability [20]. Strategies that require more participation and effort, such as cognitive reappraisal, may be most effective in regulating low-intensity emotions, whereas strategies that involve disengagement from stimuli, such as distraction, may be most effective in regulating high-intensity emotions [23,24]. In terms of perceived controllability, higher perceived controllability of one's emotional experiences should be helpful in actively regulating attempts to change one's emotional experiences [20]. When the sense of control is weakened or uncontrollable, individuals may reduce their efforts to regulate their emotions or even attempt to escape or avoid them [25].

In the field of competitive sports, important factors affecting athletic performance are often explored by comparing differences between elite and non-elite athletes. Elite shooters show relative stability in their performance across different levels of competition. They also have higher skills in emotion regulation and self-control [1]. In addition, there are significant differences in sensitivity to environmental demands and the use of regulation strategies between individuals of different genders, with females using more strategies and being more flexible in switching between strategies than males [11].

The ability model of emotion regulation posits that emotional tolerance and regulatory flexibility are two critical dimensions of emotion regulation [10,18]. The degree of emotional involvement in regulatory strategies reflects emotional tolerance [19]. Accordingly, regulatory strategies are categorized into two types. Engagement strategies are characterized by active engagement with distress or its source and encompass cognitive reappraisal, acceptance, and problem-solving. Disengagement strategies are marked by efforts to evade distressing emotions, thoughts, or situations and include expressive suppression and avoidance. The most prominent characteristic of the flexible emotional regulation process lies in the matching of emotional regulation strategies to contextual demands, namely, matching regulatory strategies to contextual demands [22]. The contextual demands of emotional regulation incorporate emotional intensity and perceived controllability. Researchers have conducted more studies on the matched emotional regulation strategies under different levels of emotional intensity [23,24], arriving at a relatively consistent conclusion: when individuals regulate low emotional intensity, engagement strategies are more effective (e.g., cognitive reappraisal), while when individuals experience high emotional intensity, disengagement strategies are more effective (e.g., distraction). Research on perceived controllability is underdeveloped compared to that on emotional intensity. Nevertheless, a high level of perceived controllability may render individuals more inclined to attempt to modify their emotional experience. In contrast, a diminished sense of control perpetuates efforts to avoid the emotional experience [25]. That is, if people believe they can alter an aversive experience, they will endeavor to do so; if they cannot change it, they will strive to escape or avoid it [26].

## 1.4. How to study flexible emotional regulation process

Emotion regulation possesses distinct characteristics. Unlike the previous studies on the relationship between emotion regulation tendencies (strategy usage preferences) and environmental adaptation [15,18,19], this research focuses on examining whether emotional regulation flexibility has a facilitating effect on sports performance. Conventional emotion regulation measurement tools can merely measure an individual's preference for emotion regulation strategies but fail to reflect the individual's potential to employ multiple diverse

emotion regulation strategies based on situational variations [21,22]. Emotion regulation tendencies and emotional regulation flexibility are two distinct characteristics of emotion regulation, and they differ significantly in terms of constructs. The original measurement tools tend to measure an individual's preference for using strategies in certain relatively stable situations. Although the core features of strategies are stable (for instance, the reappraisal strategy involves changing thoughts), the regulation context varies greatly and is dynamic. Therefore, the examination of an individual's strategy usage in the context of situational changes constitutes an investigation of short-term psychological changes. When evaluating short-term psychological changes such as relaxation, the corresponding measurement tools should give priority to considering the variability of the situation and content [27,28]. According to previous studies, the tools and items for measuring short-term psychological changes should be sensitive (capture state changes), contextual (consider the measurement background or situational variability), and individualized (measurement items adapted to different individuals).

Recent longitudinal tracking of individuals' dynamic changes in emotion regulation has gradually become an important trend in emotion regulation research [11,20–22]. This study focuses on examining the characteristic of emotional regulation flexibility. The core of emotional regulation flexibility lies in the flexible selection and use of regulatory strategies in response to situational changes, whereas traditional emotion regulation measurement tools primarily assess habitual use of emotion regulation strategies without examining situational changes. The essence of the concept of emotional regulation flexibility is to alter emotion regulation strategies flexibly according to the demands of emotion regulation (situational changes). The rules for using emotion regulation strategies themselves do not change with the context, such as cognitive reappraisal, which requires individuals to change their thoughts, altering their cognitive interpretation of different eliciting situations [29]. Based on this, the examination of emotional regulation flexibility in this study is grounded in the variability of situational changes and the selection of regulatory strategy use rules that have been widely validated in previous research, to explore the dynamic changes in athletes' emotion regulation processes.

## 1.5. Research purpose and hypothesis

The purpose of this study was to examine the influence of athletes' emotion regulation during competition on their performance using a longitudinal tracking method. This method provides a more accurate reflection of the psychological changes that occur during competition than laboratory studies or retrospective surveys. To achieve this goal, our study had two specific aims. Aim 1 was to elucidate the disparities in situational demands and the deployment of regulatory strategies among athletes varying in proficiency levels. It is hypothesized that elite athletes will exhibit lower levels of regulatory contextual demand, specifically in terms of emotion intensity and perceived controllability, when contrasted with their non-elite counterparts. Additionally, it is anticipated that elite athletes will demonstrate a distinct preference in the selection of emotion regulation strategies. The study further posited a positive correlation between the contextual demand and the typology of regulatory strategies, indicative of a dynamic emotional regulation process. This relationship suggested that high contextual demand would be associated with a predisposition towards disengagement strategies, whereas lower contextual demand would be linked to an inclination towards engagement strategies. Aim 2 was to examine the combined influence of contextual demand and regulatory strategy types on athletic performance, with a particular focus on their interplay in affecting shooting performance. It is hypothesized that differing skill levels may serve as a moderator of contextual demand and strategy types predicting shooting performance. Within the elite athlete

cohort, it is expected that a congruence between context demands and strategy types will significantly predict athletic performance. In contrast, this predictive relationship may be less pronounced among non-elite athletes. Given the potential influence of gender on the utilization of emotional regulation strategies, this variable will be controlled for within the study to ensure the accuracy and reliability of the findings.

## 2. Materials and methods

### 2.1. Participants and procedure

Participants were 60 Chinese active shooters who competed during the selection period for the 2022 World Championships in Cairo. Our final sample included 23 national and higher level shooters (8 females) and 34 first level shooters (13 females). Simulation studies of multi-level power suggest that designs with 50 Level 2 units (e.g., participants) with 14 Level 1 units (e.g., times) provide sufficient power (i.e., greater than.80) available to detect effect sizes $\geq$ .20 [30,31]. The average participants age was 23.27 (SD = 2.88) years.

Prior to data collection, we obtained verbal consent from the sports teams and coaches and written consent from each athlete participant. Prior to the initial data collection, it was explained to the athlete participants that the purpose of our study was to examine emotional regulation and athletic performance in athletes. Using ecological momentary assessment, we followed 57 athletes during competitions for a period of 6 days before the competition and 2 days during the competition (March 12, 2022 to March 20, 2022), recording emotional events, emotional intensity, perceived control, emotion regulation strategies, and shooting performance at 14 different time points, including each morning and afternoon before the competition and at the end of each competition day. Participants with more than 40% missing assessments are to be excluded [32]. Among the 60 participants, 3 athletes submitted assessments with more than 40% missing values, thus these 3 athletes were excluded from the statistical data.

All participants first learned the basic meanings of emotion regulation strategies and were able to provide appropriate examples based on their own experiences in sport to ensure that athletes fully understood the meaning of each strategy. For the experience sampling portion, participants were given a survey containing questions about the previous day once at 9:00 am and 3:00 pm for 8 consecutive days. Athletes are required to complete the assessment within 15 minutes after each daily training session and 30 minutes after each competition. Completion time will be monitored by both the questionnaire system and the investigators. (As part of a larger study, participants also completed Ecological Momentary Assessment (EMA) surveys sent twice daily.

**Ethics statement** Prior to the initiation of this research, ethical review approval has been granted by the Sports Psychology Professional Committee of Beijing Sport University. All research participants have signed written informed consent forms. For those unable to sign written consent, we provided detailed verbal explanations to ensure their full comprehension of the study content, risks, and benefits, and obtained their verbal agreement to participate in the presence of a third-party witness. We strictly adhere to ethical guidelines to safeguard the rights and interests of all participants.

### 2.2. Measures

**Contextual demands.** The assessment of emotional events and situational demands includes four items. One item examines the events that trigger the athlete's emotions("What was the event that triggered your negative emotional feelings in training today?"), another item assesses the intensity of the athlete's emotions("Assess the intensity of this negative emotional feeling"), and two items assess controllability: emotional controllability ("To

what extent do you feel you were in control of your emotional response?") and situational controllability ("To what extent were you in control of what happened to you?"). Items are scored on a 5-point scale ranging from 1="not at all" to 5 = "very much".

**Regulatory strategies.** According to the ability model of emotion regulation, the core characteristics of the six regulatory strategies are relatively stable. In this study, the items for the six emotion regulation strategies were derived from existing measurement tools, extracting the key characteristics of each strategy. Acceptance involves merely noticing without judgment [30]. Problem-solving refers to identifying specific and concrete methods to deal with current emotions [33]. Cognitive reappraisal involves individuals changing their thoughts about a situation to alter their emotions. Expressive suppression entails inhibiting the emotions felt and refraining from expressing them to achieve emotional regulation [34]. Rumination is characterized by repetitive or immersive experiences or thoughts about certain emotions [35]. Avoidance encompasses a variety of strategies to escape, control, or suppress unwanted thoughts, emotions, and sensations [36]. To assess the emotional regulation process of athletes within a relatively short timeframe, brief measurements are necessary to avoid excessively prolonging the testing period. Although the effectiveness of emotion regulation strategies may be influenced by context, the rules governing the use of these strategies do not change with context.

Participants were asked, "How do you cope with the events that trigger the emotional experiences mentioned above? Which of the following strategies did you choose to cope with them?"

*Cognitive reappraisal* was measured with an item "I tried to think about the situation differently." *Acceptance* was measured with "I tried to notice my thoughts and feelings without thinking of them as good or bad." *Problem-solving* was measured with "I tried to come up with a specific strategy to work through my anxiety." *Expressive suppression* was measured with "I made sure not to express what I was feeling." *Rumination* was measured with "I kept think-ing about and dwelling on my anxiety." *Avoidance* was measured with an item that read, "I tried to avoid thinking about what happened" (referred to as "avoidance" in our study), and experiential avoidance of anxiety was measured with an item that read, "I tried to control my anxiety-related feelings or thoughts," which was drawn from a validated state measure of expe-riential avoidance [36]. Use of each strategy was rated on a 5-point scale ranging from 1="not used at all" to 5 = "used completely". These items reflect the core content of the six strategies and do not alter the content structure of the emotion regulation strategies, which is in line with recent emotion regulation (ER) research that has employed experience-sampling meth-odology (ESM; real-time assessment at multiple time points in different contexts) [37,38].

**Shooting performance.** Comprehensive evaluation of shooting performance is conducted through shooting accuracy and shooting stability, as demonstrated in studies by Ihalainen et al [39,40]. Shooting accuracy is measured by the radial distance from the center of the target to the hit point, which serves as a method to assess the shooting precision score. The shooting ring value (Score) accurately reflects shooting accuracy, ranging from 0 to 10.9. A higher shooting ring value indicates a shorter distance from the target center to the hit point, representing enhanced shooting accuracy and superior shooting performance [41]. Shooting stability primarily refers to the consistency and stability of the athlete's shooting execution. It is commonly evaluated using the percentage of good ten-ring hits, which signifies the proportion of shots with a ten-ring value out of the total number of shots taken(The scoring rings on a shooting target are divided from the outer 6-ring to the central 10-ring, with each whole number ring further divided into 10 parts, ranging from 6.0 to 6.9, and so on, up to the central 10-ring which ranges from 10.0 to 10.9). A higher percentage of good ten-ring hits indicates better shooting stability [42]. During training and competitions,

electronic target systems are utilized to record both the shooting Score and the percentage of good ten-ring hits.

## 2.3. Analysis

SPSS version 22.0 was used to conduct Difference Test, and R version 3.6.1 was used to conduct Multi-level Linear Model analysis. A two-level model was constructed with the dependent variables being the average shooting score and the proportion of good ten rings. Time-level (Level 1) linear mixed-effects models were conducted with lme4 [43]. Participants were treated as random effects in all mixed-effects models with a random intercept and no random slopes. Level 1 predictors were person-mean centered. Different levels of athletic ability was the second predictor. Full statistical information for each model was created using sjPlot [44]. Two effect sizes were calculated: marginal $R^2$ m (i.e., the proportion of the total variance explained by the fixed effects) and conditional $R^2$ c (i.e., the proportion of the total variance explained by both fixed and random effects) [45].

## 3. Results

### 3.1. Descriptive statistics

Descriptive statistics for all variables are presented in Table 1. There were significant differences in emotional intensity between athletes of different levels, with first-level athletes having significantly higher emotional intensity than elite-level athletes (t = 4.39, $p < 0.001$), but no significant difference in controllability. First-level athletes used more regulation strategies than national athletes on these two types of emotion regulation strategies, the First-level athletes used both types of strategies significantly more than the Elite-level athletes. In terms of shooting performance, first-level athletes had significantly lower good ten-ring ratios than national athletes.

Emotional regulation flexibility is primarily characterized by individuals' selection and application of regulatory strategies that are congruent with contextual demand, particularly in terms of emotional intensity and perceived controllability. Drawing from previous research on the relationship between contextual demands and the choice of emotional regulation strategies [19,23–25], contextual demand are composed of emotion intensity and controllability. When emotional intensity increases and controllability decreases, the demands on

**Table 1. Descriptive statistics.**

|  | Level | | |
|---|---|---|---|
|  | **First-level** | **Elite-level** | **t** |
| Emotional Intensity | 5.59(1.91) | 4.8(1.72) | 4.39*** |
| Controllability | 4.91(2.30) | 5.34(2.26) | −1.90 |
| Contextual demand | 7.50(2.06) | 6.92(1.44) | 3.213** |
| Engagement strategy | 8.07(3.53) | 6.47(3.02) | 4.92*** |
| Disengagement strategy | 7.40(2.99) | 6.02(2.24) | 5.21*** |
| Regulatory strategy type | −0.68(4.03) | −0.45(2.70) | 0.656 |
| Average Ring Value | 9.92(0.33) | 9.98(0.28) | −1.95 |
| Good Ten-ring Ratios | 0.33(0.07) | 0.40(0.07) | −9.91*** |

Note:

*p < 0.05,

**p < 0.01,

***p < 0.001, the same below.

the individual's context rise, prompting the selection of different strategies for regulation. Previous studies have separately examined the relationship between emotional intensity and controllability with the choice of strategy types, but these two variables collectively reflect contextual demands, and they are interdependent. When emotional intensity is high, controllability tends to be poor [20,26]. Therefore, after calculating the scores for the direction of controllability and adding them to the emotional intensity scores, the combined total reflects the contextual demands. The higher the total score, the greater the emotional intensity and uncontrollability, and the higher the need for regulation. The results are presented in Table 1, which shows that elite athletes have significantly lower regulatory demands compared to first-level athletes.

To more accurately assess the extent to which individuals select regulatory strategies congruent with their contextual demand, we calculated the individual's daily propensity for choosing these strategies by determining the difference between disengagement strategy scores and engagement strategy scores. The findings indicate no significant differences in the types of regulatory strategies employed by elite athletes compared to first-level athletes. The results are shown in Table 1, a negative value for the regulation strategy type suggests that both groups predominantly favor engagement strategies, demonstrating a more proactive approach to adjusting their emotional experiences during training and competition.

Correlations for all variables are shown in Table 2. The correlation analysis between the contextual demand and regulatory strategy type showed that when the contextual demand is stronger (emotion intensity and perceived uncontrollability), athletes tend to choosed disengagement strategies (e.g., distraction); while when the contextual demand is lower, athletes tend to choose engagement strategies. The two indicators for evaluating athletic performance (Average Ring Value and Good Ten-ring Ratios) were not significantly correlated with the contextual demand and regulatory strategy type, because athletic performance is related to the entire flexible emotional regulation process rather than any specific aspect of it. The ICC showed that 82% and 56% of the variation in the regulatory strategy type, respectively, came from between individuals. Therefore, it is reasonable to further analyze the effects of between-group and within-group factors on shooting performance using a multilevel structural model.

## 3.2. Hierarchical analysis on the influence of emotion regulation on shooting performance

Emotion regulation flexibility involves aligning regulatory strategies with contextual demands, referred to as situation-strategy fit (SSF). Based on current research-derived matching patterns (employing disengagement strategies when contextual demands are high and engagement strategies when contextual demands are low), the daily SSF of shooters can be calculated. This calculation involves multiplying the contextual demands by the regulatory strategies, providing a sensitive reflection of shooters' flexibility in using matching strategies according to the level of contextual demands.

**Table 2. Correlation between contextual demand, regulatory strategy type and shooting performance.**

|  | ICC | 1 | 2 | 3 |
|---|---|---|---|---|
| 1. Contextual demand | 0.65 |  |  |  |
| 2. Regulatory strategy type | 0.31 | 0.18** |  |  |
| 3. Average Ring Value | 0.23 | 0.07 | −0.40** |  |
| 4. Good Ten-ring Ratios | 0.36 | 0.17** | −0.12* | 0.46** |

The aim of this study is to investigate whether shooting performance is more influenced by dynamic individual factors during the athlete's competitive period or by stable within-person variables. To achieve this, a multilevel linear model was used to analyze longitudinal data over a period of 8 days. We first ran main effects models examining how psituation-strategy fit (SSF) predicted the shooting performance (see Table 3). Situation-strategy fit was entered as predictors in separate models, and average ring value and good ten-ring ratios was predicted separately.

We then examined athletic level as a moderator of contextual demands and regulatory strategy type predicting shooting performance. In these models, shooting performance is the outcome, negative emotion contextual demand and tegulatory strategy type are the Level 1 predictors, and athletic level is a Level 2 moderator:

$$\text{Day} - \text{level} : Y_{ij} \text{ Performance } = \beta_{0j} + \beta_{1j} \text{ Situation} - \text{strategy fit} + r_{ij}$$
$$\text{Person} - \text{level intercept} : \beta_{0j} = \gamma_{00} + \gamma_{01} \text{ Level} + \mu_{0j}$$
$$\text{Person} - \text{level slope} : \beta_{1j} = \gamma_{10} + \gamma_{11} \text{ Level} + u_{1j}$$

The Level * Situation-strategy fit interaction terms represent a test of group differences; significant values indicate that situation-strategy fit predict performance for shooters with first-level than elite-level. Simple slopes represent the relationship between situation-strategy fit and shooting performance in each athletic level (i.e., First-level or elite-level). Full models are provided in Table 4.

In main effects models(see Table 4), higher situation-strategy fit was associated with higher good ten-ring ratios. Thus, the more situation-strategy fit shooters got, the greater they got shooting performance. Examination of simple slopes suggests this was true for elite-level athletes and first-level athletes. For good ten-ring ratios, there was a significant cross-level interaction, meaning that the strength of the relationship differed between atheltic levels participants. Specifically, situation-strategy fit and good ten-ring ratios were positively correlated in both groups, but the strength of this relationship was stronger for elite-level athletes than first-level athletes. In main effects models, higher situation-strategy fit was associated with average ring value. Examination of simple slopes suggests that for first-level athletes, situation-strategy fit was unrelated to average ring value. This differed from elite-level athletes, where higher situation-strategy fit was associated with average ring value.

## 4. Discussion

### 4.1. Differences in situational demands and the deployment of regulatory strategies among athletes of different level

First-level athletes experience greater emotional intensity and have lower controllability. Compared to elite-level athletes, they use emotional regulation strategies more frequently. First-level athletes have significantly lower shooting scores (Average Ring Value) and

**Table 3. Main effects of situation-strategy fit on average ring value and good ten-ring ratios.**

| Predictor | Average Ring Value | | | | Good Ten-ring Ratios | | | |
|---|---|---|---|---|---|---|---|---|
| | b | CI | t | p | b | CI | t | p |
| Intercept | 1.15 | 1.37, 1.78 | 25.02 | <0.001 | 1.28 | 1.51, 1.93 | 23.18 | <0.001 |
| Situation-strategy fit | 0.26 | 0.11, 0.37 | 6.01 | <0.001 | 0.20 | 0.14, 0.47 | 8.21 | <0.001 |
| Observations | 786 | | | | 786 | | | |
| Marginal R²/Conditional R² | 0.041/0.298 | | | | 0.036/0.364 | | | |

Table 4. Multilevel models: Athletic level moderation of situation-strategy fit predicting shooting performance.

| Predictor | Average Ring Value | | | | Good Ten-ring Ratios | | | |
|---|---|---|---|---|---|---|---|---|
| | b | CI | t | p | b | CI | t | p |
| Intercept | 1.26 | 1.16, 1.48 | 26.19 | <0.001 | 1.28 | 1.51, 1.93 | 23.18 | <0.001 |
| Situation-strategy fit | 0.19 | 0.09, 0.41 | 6.83 | <0.001 | 0.20 | 0.14, 0.47 | 8.21 | <0.001 |
| Athletic Level | 0.42 | 0.31, 0.75 | 7.11 | <0.001 | 0.25 | 0.08, 0.44 | 3.01 | <0.001 |
| Situation-strategy fit × Athletic Level | −0.03 | −0.12, 0.37 | −0.41 | 0.518 | 0.22 | 0.17, 0.69 | 3.12 | 0.002 |
| Observations | 786 | | | | 786 | | | |
| Marginal $R^2$/Conditional $R^2$ | 0.115/0.351 | | | | 0.043/0.325 | | | |
| Simple Sloppe for Elite-level athletes | 0.08 | 0.01, 0.16 | 2.11 | 0.034 | 0.17 | 0.08, 0.27 | 3.87 | <0.001 |
| Simple Sloppe for First-level athletes | −0.04 | −0.03, 0.12 | −1.28 | 0.201 | 0.12 | 0.09, 0.21 | 3.26 | <0.001 |

percentages of perfect scores (Good Ten-ring Ratios), which may be related to their differences in emotional regulation. Elite athletes exhibit significantly lower regulatory demands compared to first-level athletes. However, there is no significant disparity in the selection of regulatory strategy types between elite-level athletes and first-level athletes. Both groups employ engagement strategies more frequently, meaning they adjust their emotional feelings more diligently and actively during training and competitions. This is associated with the fact that shooting athletes place greater emphasis on emotion regulation. The shooting process demands high levels of focus from athletes and requires a lower level of emotional arousal [3,6], thus they tend to undertake emotional adjustments more proactively. The disengagement strategy scores suggest that low-level athletes utilize more strategies such as distraction and suppression of expression. Nevertheless, these disengagement strategies are not always detrimental, as elite-level athletes also employ a considerable number of disengagement strategies. When the contextual demand is stronger (emotional intensity and perceived uncontrollability), athletes tend to opt for disengagement strategies (e.g., distraction). This is in alignment with the findings of current research on the characteristics of emotional regulation flexibility [19,23–25]. When the background demand is high, that is, when the emotional intensity is high or the perceived controllability is low, an individual's cognitive processing resources are in a strained state. Disengagement strategies have low demands on cognitive resources, and individuals' utilization of these strategies can more effectively mitigate the detrimental impacts of negative emotions [20,42,46].

## 4.2. Hierarchical analysis of the influence emotional regulation on shooting performance

At the day-level, we explored the relationship between situation-strategy fit and shooting performance, and examined the predictive effect of situation-strategy fit on shooting performance at the individual level. This relationship was fully validated among elite-level athletes, but among first-level athletes, only the significant prediction of situation-strategy fit on good ten-ring ratios was verified. Situation-strategy fit is a core characteristic of emotional regulation flexibility [22,23]. Previous studies have demonstrated the match by examining the relationship between situational demands and strategy use. Situational demands often encompass emotional intensity and perceived controllability [20,26]. Individuals' perceptions of controllability are a key factor in the emotional regulation process and often influence their choice of emotional regulation strategies. When the level of controllability is higher, individuals tend to choose strategies such as suppression and avoidance [20,46]. In contrast

to suppressing emotional experiences, suppressing physiological responses to emotions can improve performance [17]. The emotional experience of shooting athletes during competition is often influenced by their performance and physiological changes [1,6]. Studies have shown that athletes who use disengagement strategies more often have lower shooting performance, possibly because they use more disengagement strategies in the emotional experience component of emotions, which alleviates their subjective emotional experience, but their heart rate, breathing, and other physiological indicators remain high, affecting their stability during competition [3,6]. Emotions are a key factor in influencing athletic performance, and different emotional intensities have different effects. When the intensity of negative emotions is low, the emotion regulation strategy of expressive suppression, by avoiding or deliberately ignoring bodily sensations, reduces over-exertion due to tension while improving the accuracy of the shot [17]. The emotional regulation process is a crucial factor influencing athletic performance, broadly affecting the execution of movements. Relying solely on individual factors within the emotional regulation process (e.g., emotions, emotional regulation strategies) can no longer accurately reflect the effectiveness of emotional regulation in complex sports situations [1,11]. It is necessary to integrate these factors to predict athletic performance. In this study, the relationship between situation-strategy fit and athletic performance, as we hypothesized, showed that the higher the fit, the better the athletic performance. The fit between situation-strategy is a core characteristic of emotional regulation flexibility [22,23]. Previous research has demonstrated the match by exploring the relationship between situational demands and strategy use. Situational demands often include emotional intensity and perceived controllability [20,26].

The emotional regulation process is a critical factor influencing athletic performance, broadly affecting the execution of movements. Relying solely on individual factors within the emotional regulation process (e.g., emotions, emotional regulation strategies) can no longer accurately reflect the effectiveness of emotional regulation in complex sports situations [1,11]. It is necessary to integrate these factors to predict athletic performance. In this study, the relationship between situation-strategy fit and athletic performance, as we hypothesized, showed that the higher the fit, the better the athletic performance. The fit between situation-strategy is a core characteristic of emotional regulation flexibility [22,23]. Previous research has demonstrated the match by examining the relationship between situational demands and strategy use. Situational demands often include emotional intensity and perceived controllability [20,26].

Adding a second layer of variables, athletic level comprehensively reflects the athlete's tactical and technical level, including the level of psychological skills. Studies have shown that elite athletes have higher psychological adjustment capabilities [8,11]. Therefore, we examined whether athletes at two levels differ in the prediction of emotion regulation on athletic performance. The results were as expected, with elite-level athletes' situation-strategy fit being a stronger predictor of two indicators of shooting performance. Compared to first-level athletes, elite-level athletes have higher average ring values and good ten-ring ratios when using engagement strategies under lower contextual demand and using disengagement strategies under higher contextual demand. First-level athletes may have a mismatch between perceived situational demands and regulation strategies. This may be because first-level athletes do not necessarily use regulation strategies when they perceive situational demands [20,47]. In addition, engagement strategies are more effective at regulating low-intensity emotions but require a greater allocation of cognitive resources [24]. The execution of shooting movements requires the shooter to concentrate a large amount of attention resources [1,12,24], and first-level shooters may not have the ability to allocate attention resources well. In subsequent data, we further compared the situation-strategy fit of the two groups of athletes and found a marginally significant difference, with elite-level athletes being higher.

Thus, the emotional regulation process has a significant impact on shooting sports performance, and the ability to use matching strategies in situational demands affects the effectiveness of emotional regulation. Situation-strategy fit can serve as an important predictive factor for athletic performance. This suggests that improving athletes' flexible use of emotion regulation strategies may help improve shooting performance.

## 5. Limitations

Shooting athletes select and use different regulation strategies according to situational demands. This study attempts to elucidate the impact of emotional regulation flexibility on shooting performance by comparing emotional regulation differences between first-level and elite-level athletes. The situation-strategy fit of elite-level athletes is higher, and their prediction of shooting performance is stronger. However, first-level athletes perform unsatisfactorily in situation-strategy fit. The disparity in their emotional regulation contextual demands and strategies may be a crucial reason for the variance in their training and competition performance. By promptly recalling and recording emotions and regulation processes after tasks, this study more realistically reflects the impact of emotional regulation processes on athletic performance during athletic competition with higher ecological validity. However, athletes' retrospective assessment of emotional induction, intensity, and controllability is an evaluation of the overall situation over a short period, not the emotional intensity and controllability that occur in real-time during training and competitions. This also makes it difficult to examine the dynamic changes in athletes' emotional regulation strategies and situational demands over this period. This is an inherent limitation of survey research, which is the inability to record more detailed data more densely. Research that relies solely on retrospective self-report data demands that athletes a) thoroughly remember their emotions and strategies, and b) be able to clearly identify the strategies they have employed, which is a challenging task. Although we have endeavored to minimize this bias by employing immediate post-event recall methods during our study, the inherent limitations of retrospective surveys in accurately reflecting real-time dynamic changes are inescapable.

The dynamic variations in emotional regulation strategies, which are context-dependent and not accurately detectable by currently recognized measurement tools, represent a challenge. The measurement tool employed in this study was developed by integrating core characteristics of strategies into new items based on the original Emotional Regulation Strategy Scale, which may limit the measurement validity. Future research could further validate the measurement validity in such diary studies by selecting different validity indicators. Additionally, the physiological changes during emotional regulation and the shooting process, such as heart rate and EEG, could not be effectively collected due to restrictions imposed by training and competition conditions, which do not permit athletes to wear any monitoring devices. Using subjective assessment methods to examine the induced emotions (emotional intensity) in individuals is not sufficiently clear and may be subject to bias. This may result in a lack of physiological evidence to support the data analysis. The computational method for emotion regulation flexibility, which was first used after a thorough review and respect for consistent research findings, can accurately reflect the core features of emotion regulation flexibility, namely, the matching between context and strategy. Meanwhile, further validation from different research backgrounds is needed. In the future, more rigorous experimental studies will be conducted to further explore whether this matching between context and strategy is universally present.

## 6. Perspects

Further experimental studies are needed to accurately reflect which emotional regulation strategies athletes choose under different emotional intensities and controllability, and to

reveal which key link in the emotional regulation process has a greater impact on behavioral performance. In addition, the effects of different emotional regulation strategies on emotional experience, physiological components, and cognitive components are not entirely similar. Future studies should explore this further in order to provide more specific guidance for emotional regulation that enhances athletic performance.

## 7. Conclusion

Emotional regulation is often considered a crucial component of athletes' psychological abilities [8,12] Beatty, 2019;. Athletes often employ a variety of emotion regulation strategies to modulate intense emotional fluctuations, thereby promoting competitive performance [9]. The complexity and variability of sports situations demand that athletes flexibly use different regulation strategies according to situational needs. The primary purpose of athletes' emotion regulation is to enhance athletic performance. Therefore, exploring the relationship between flexible emotion regulation processes and sports performance can provide directional guidance for emotional regulation training.

Our results lead to the following conclusions. First-level athletes have a higher contextual demand for emotion regulation and tend to use disengagement strategies more frequently to regulate emotions. The situation-strategy fit is characterized by the use of disengagement strategies in high situational demand and engagement strategies in low situational demand. This match exerts different impacts on shooting performance between the two groups. Elite-level athletes have higher average ring values and good ten-ring ratios when using engagement strategies under lower contextual demand and using disengagement strategies under higher contextual demand. First-level athletes may have a mismatch between perceived situational demands and regulation strategies. Therefore, enhancing flexible emotional regulation training that improves situation-strategy fit may be beneficial for enhancing sports performance.

## Supporting information

**S1 Data. Research data set.**
(CSV)

**S2 Data. An introduction to the naming of variables in the study.**
(CSV)

## Acknowledgments

We thank Cai Yalin, Peng Duobao, and Liu Bin for their instrumental help with data collection.

## Author contributions

**Conceptualization:** Zhou Lulu, Su Hua.

**Data curation:** Zhou Lulu, Su Hua.

**Formal analysis:** Liu Huimin.

**Funding acquisition:** Zhou Lulu.

**Investigation:** Su Hua.

**Methodology:** Zhou Lulu, Liu Huimin, Su Hua.

**Resources:** Liu Huimin.

**Validation:** Liu Huimin.

**Writing – original draft:** Zhou Lulu.

**Writing – review & editing:** Zhou Lulu, Su Hua.

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
