## [Decision Letter · Decision Letter 0]

16 Sep 2024

PONE-D-24-25633Effect of Emotional Regulation on Performance of Shooters during Competition: an Ecological Momentary  Assessment StudyPLOS ONE

Dear Dr. Hua,

Thank you for submitting your manuscript to PLOS ONE. After careful consideration, we feel that it has merit but does not fully meet PLOS ONE’s publication criteria as it currently stands. Therefore, we invite you to submit a revised version of the manuscript that addresses the points raised during the review process. Editor comment. I was fortunate to receive feedback from three reviewers, all of whom are top experts. They found the paper well-written and the topic interesting, but there are some important issues to address. R1 mentioned the need for more clarity, especially in the abstract and the way you interpret your findings, along with how you operationalise "flexibility" in emotion regulation. R1 also suggested adding some further analyses to deepen your insights. R2 pointed out the need for more transparency around your statistical methods, including clearer explanations of the software and models you used, and asked for more detailed reporting of the regression equations and coefficients. The most significant concerns came from R3, who raised concerns about the validity of the items used to measure emotion regulation strategies (especially whether single items can be used from the context of the ensemble of the questionnaire) and noted that the time factor was not incorporated in your longitudinal analysis, which R3 argues is essential. These are critical points that will require careful revision. I thereby invite you to submit a revised manuscript that includes a point-by-point response to all of the reviewers' comments.

We look forward to receiving your revised manuscript.

Kind regards,

Michael B. Steinborn, PhD

Section Editor

PLOS ONE

Journal Requirements:

"Psychological Capacity Evaluation and Talent Selection Research of High-level Shooting and Archery Athletes" (2024JT07) . "

4. For studies involving third-party data, we encourage authors to share any data specific to their analyses that they can legally distribute. PLOS recognizes, however, that authors may be using third-party data they do not have the rights to share. When third-party data cannot be publicly shared, authors must provide all information necessary for interested researchers to apply to gain access to the data. (https://journals.plos.org/plosone/s/data-availability#loc-acceptable-data-access-restrictions)

 a) A description of the data set and the third-party source

 b) If applicable, verification of permission to use the data set

 c) Confirmation of whether the authors received any special privileges in accessing the data that other researchers would not have

 d) All necessary contact information others would need to apply to gain access to the data

Reviewers' comments:

Reviewer's Responses to Questions

**Comments to the Author**

1. Is the manuscript technically sound, and do the data support the conclusions?

Reviewer #1: Partly

Reviewer #2: Partly

Reviewer #3: No

2. Has the statistical analysis been performed appropriately and rigorously? 

Reviewer #1: Yes

Reviewer #2: Yes

Reviewer #3: No

3. Have the authors made all data underlying the findings in their manuscript fully available?

Reviewer #1: No

Reviewer #2: No

Reviewer #3: No

4. Is the manuscript presented in an intelligible fashion and written in standard English?

Reviewer #1: No

Reviewer #2: Yes

Reviewer #3: Yes

5. Review Comments to the Author

Reviewer #1: In this study, the authors assessed emotion regulation strategies in female and male shooters at both first-level and national-level using longitudinal ecological momentary assessment (EMA). They aimed to predict the impact of these strategies on shooting performance. The results revealed differences in emotion regulation based on gender and skill level, and their relation to shooting performance.

The study is compelling, with the use of EMA enhancing the ecological validity of the findings. However, the manuscript lacks clarity in several sections. Additional analyses could better address the research questions posed by the authors. While the discussion provides sufficient explanations for many results, it overlooks some key points. Therefore, I recommend a major revision of the manuscript.

See below for detailed feedback Unfortunately, the authors did not provide page or line numbers. Therefore, I will refer to section headings in my feedback:

1. In the Abstract, you state: “Inaccurate perception of situational demands and difficulty in using engagement strategies are the main emotion regulation processes that affect athletic performance.” However, this conclusion appears to be unsupported by the presented results. The study does not seem to assess the accuracy of athletes' perception of situational demands. Moreover, this statement does not align with the primary findings of your paper, which focus on the differences in emotion regulation strategies between male and female athletes, as well as between first-level and national-level competitors. Consider revising the Abstract to more accurately reflect the key points and results of your research.

2. You refer to the flexibility in the usage of emotion regulation strategies. However, it is unclear which results of your study indicate high or low flexibility. Is it the sum score of the usage of all strategies? Conclusions regarding overall flexibility would require calculating a variable that numerically captures this construct. I am not sure if your data can provide information on emotional flexibility. Additionally, interpreting this in absolute terms would be difficult, as you would require a non-athlete control group to determine if your sample is indeed less flexible in their usage of emotional control strategies. Therefore, I suggest that you only interpret differences between your subgroups (gender and skill level). If you cannot operationalize the flexibility in emotional regulation, please avoid jumping to conclusions (e.g., “Athletes had low flexibility in emotion regulation …” [Abstract], “Low-level athletes have lower emotional regulation flexibility compared to high-level athletes, and they demonstrate inflexibility in both situational demands and specific” [4. Limitations], etc.).

3. You use some terminology from shooting sports that laypeople (such as myself) might find difficult to understand. Specifically, please elaborate on the outcomes of your models:

- What is the “proportion of good ten rings” [2.4 Analysis] exactly?

- What is meant by “speed of grip force” [Section 1.1] and how is it related to shooting performance?

- What exactly is meant by “slow the production of force” [Section 3.2] and again, how does this relate to performance?

Additionally, in Section 1.3 you refer to “rapidly changing competitive sports situations.” What do you mean by that? How is this related to shooting (i.e., what are the changing situations in a shooting competition)?

4. Regarding your power analysis: If I am not mistaken, gPower is not able to perform power analyses for multilevel linear models. It appears that you computed a power analysis for a linear multiple regression with 4 predictors, resulting in the reported detectable effect sizes of f²=0.20 (I assume that you refer to the effect size parameter f², please make so explicitly). However, a power analysis based on a (non-hierarchical) linear regression is not suitable for a multilevel analysis.

Please conduct a suitable power analysis (for example, using a simulation-based approach) or use another method of justification for your sample size. You may find the following resource helpful:

https://online.ucpress.edu/collabra/article/8/1/33267/120491/Sample-Size-Justification
https://shiny.ieis.tue.nl/sample_size_justification/

5. Please report how many surveys were completed and the extent of dropout you experienced. Additionally, specify if you had any exclusion criteria for the data, such as excluding participants with a survey completion rate below a certain threshold.

6. Please provide all survey questions asked in the 2.2 Measures section. The questions for the triggering event and the emotional intensity are missing.

7. Please report the nationalities of the athletes.

8. The main effect models are interesting, but what about Level 1 interactions? At least two-way interactions might yield interesting results, especially in light of the literature, which has shown that engagement strategies are most effective in low-intensity emotional contexts, while disengagement strategies are more effective in high-intensity emotional contexts. You make several remarks regarding such an interaction:

- “Strategies that require more participation and effort, such as cognitive reappraisal, may be most effective in regulating low-intensity emotions, whereas strategies that involve disengagement from stimuli, such as distraction, may be most effective in regulating high-intensity emotions” [Section 1.3]

-“In addition, engagement strategies are more effective at regulating low-intensity emotions, but require a greater allocation of cognitive resources” [Section 3.2]

Exploring these interactions in your models could provide a deeper understanding of how emotion regulation strategies function under varying emotional intensities. I would thus suggest to include a fourth model which encapsulates all two-way interactions of level 1 variables. I would thus suggest including a fourth model that encapsulates all two-way interactions of Level 1 variables. This additional analysis could provide a deeper understanding of how different emotion regulation strategies function under varying emotional intensities and contexts.

9. To strengthen the analysis, I suggest performing a model comparison analysis. Since Model 1 is not nested within Model 2, using an Information Criterion like the Akaike Information Criterion (AIC) and/or the Bayesian Information Criterion (BIC) would be appropriate. These criteria allow for the comparison of non-nested models by evaluating the trade-off between model fit and complexity.

10. At the moment, you only report the random intercept. Could you also report the random effects of gender and (skill-)level?

11. I noticed some inconsistencies in your descriptives, specifically in Table 1: The average ring value is approximately 10, or exactly 10.00 for male athletes. Isn’t this the highest possible value achievable, meaning that they shot the center of the target almost every time (I apologize, I am a layperson regarding shooting)? Is this correct? Could there be a ceiling effect in accuracy? Additionally, how does this align with the ten-ring ratio being approximately 35%? Shouldn’t this be close to 100% if the average ring value was 10? Please clarify these points and explain the potential discrepancy between the average ring value and the ten-ring ratio.

Also, please add some context regarding the highest achievable score in the average ring value. If these numbers are indeed correct, you should discuss these near-perfect performances of the athletes and the possible ceiling effects. This context will help readers better understand the skill level of the athletes and the implications for your study's findings.

12. “National-level athletes have higher performance levels and significant differences in emotional experience and use of regulatory strategies, which may account for the significant differences in shooting performance.” [Section 3.1] This conclusion is likely confounded with athlete experience and habituation. National-level athletes have probably competed in many competitions and might have habituated to the emotional distress experienced during competition. Their experience might also be a reason for the better performance, raising the question of whether emotion regulation strategies are indeed the primary reason behind their greater success. Please discuss these results more carefully.

13. The discussion of gender differences in Section 3.1 is somewhat brief. Please elaborate further on the differing results for females compared to males. What are the implications of these findings, and why do these differences exist?

14. “However, the intensity and controllability of emotions are averaged, making it difficult to examine the variability and degree of adaptation of emotional regulation strategies under different situational demands.” [Section 4. Limitations] I do not understand this limitation. You used a hierarchical model for your analysis, so you did not average over intensity and controllability of emotions, correct?

15. Please discuss the limitations in more detail. For instance, you relied solely on retrospective self-reported data. This requires athletes to a) remember their emotions and strategies thoroughly and b) be able to clearly identify their used strategy, a task that is arguably even hard for mental health practitioners. Also, you did not assess physiological data, which might be crucial to understand in light of the shooting performance. Additionally, the lack of preregistration is a significant limitation, as preregistration is considered part of good scientific practice. Please reflect on these and any other limitations in your study.

16. A Conclusion section at the end of your discussion would round off the paper nicely.

17. The manuscript would benefit from increased clarity in certain sections. For instance:

• “Expressive suppression is the most commonly used strategy by athletes in intense competition with limited time （Beatty & Janelle，2016）, but the effective time is short, and the negative experience caused by high intensity may disrupt the movement structure.” [Section 1.1] Could you please elaborate on the statement “and the negative experience caused by high intensity may disrupt the movement structure”? This phrase would benefit from further explanation to clarify its meaning and implications

• “The prerequisite for effective emotion regulation strategies is the context that triggers emotional changes.” [Section 1.1] Unfortunately, I do not understand what you mean. Could you clarify this?

• “Previous studies have focused on exploring regulation strategies or patterns that are applicable to sport situations, and compared to life situations, sport situations are constantly changing, and the applicability of specific regulation strategies is challenged.” [Section 1.1] What do you mean by “compared to life situations, sport situations are constantly changing”? It's debatable whether sport situations are inherently more dynamic than everyday life scenarios, which can also be highly variable and unpredictable. Could you please elaborate on this comparison and provide evidence or specific examples to support your assertion?

• “The ability-based model suggests that emotional clarity, emotional tolerance, and emotional regulation flexibility are the potential for individuals to regulate emotions.” [Section 1.2] Please clarify what you mean by “are the potential for individuals to regulate emotions”

• “Rumination (disengagement with a tendency to avoid difficulties) is also classified as a disengagement strategy （Goodman，2021)”I don’t understand what is meant by “disengagement with a tendency to avoid difficulties”, please clarify. [Section 1.2]

• “However, in rapidly changing competitive sports situations, the effects of specific emotion regulation strategies are not stable, and athletes need to flexibly choose and use different strategies based on changes in the situation to achieve good results.” [Section 1.3] What do you mean by “rapidly changing competitive sports situations”. How does this specifically relate to shooting?

• “…the second-level variables were individual differences among participants (level, gender)” At this point in the manuscript, it was unclear to me, what “level” means. Maybe re-label it to “skill-level” or something more comprehendible.

• “Model 2: A regression model with mean as the outcome variable is used to conduct […]” [Section 3.2] Please clarify which mean (of what?) is used as an outcome

18. Your manuscript sometimes uses inconsistent terminology which impairs flow and understandability:

• You define “the proportion of good ten rings” [2.4 Analysis] as one of the outcomes. However, you refer to this outcome as “Ten-ring Rations” [Table 1 to Table 3], or “proportion of perfect scores” [Section 3.2].

• You refer to one type of emotion regulation strategies as “engagement strategies” but you refer to this as “participation strategy” in Section 3.2

19. Some statements in your manuscript require literature references:

• “It can be seen that the effect of specific emotion regulation strategies is not stable.” [Section 1.1]

• “Regulation strategies that successfully regulate emotions or tolerate aversive emotions are better suited for athletes to achieve their goal of enhancing sport performance than regulation strategies that are more adaptable to individual psychological health development.” [Section 1.2]

• “Compared to male athletes, female athletes use six emotional regulation strategies more frequently, which is consistent with findings from studies of college students that show stable gender differences in the emotional regulation process.” [Section 3.1]

• “This may be because females have more complex emotions than males.” [Section 3.2] This is a strong claim and urgently needs a reference!

• “Women are more sensitive to situational demands and use more regulatory strategies.” [Section 3.2]

• “In addition, engagement strategies are more effective at regulating low-intensity emotions, but require a greater allocation of cognitive resources.” [Section 3.2]

20. There seems to be a citation error in Section 3.2:

• “In contrast to suppressing emotional experiences, suppressing physiological responses to emotions can improve performance（18）.” (no author or year provided)

Reviewer #2: The authors aimed to examine the effect of emotion regulation on

shooting performance in shooting athletes. I have a few questions regarding the methodology.

1. The authors mentioned that they used statistical software for data processing. However, the explanation lacked detail, such as which menus or features were utilized. This level of detail was necessary to allow other researchers to replicate the study described in this manuscript. As a suggestion, researchers could consider using statistical analysis with programming languages to facilitate easier replication.

2. In the manuscript, the authors mentioned the term 'ACC' but never explained its meaning. This could make it difficult for general readers to understand its significance.

3. The authors developed three regression models. In the first model, a two-level approach was used. In the second model, no predictors were included at the second level. Why did the authors choose not to use a standard multiple regression model? It appears there is no significant difference. Please explain.

4. In the second model, the authors coded gender as 1 for males and 2 for females. Why did not the authors use 0 for males and 1 for females, given that gender was nominal and not ordinal? Please explain.

5. The researchers should consider providing the equation form of all the models generated, including the value of each coefficient and intercept. This will allow readers to visually assess the influence of each predictor on the target.

The manuscript has the potential to be published. However, certain aspects, particularly in the methodology, require a more detailed explanation regarding how the conclusions were reached.

Reviewer #3: Introduction

The introduction is written quite well.

Method

Unfortunately, I regret to say that the work contains some serious problems.

- The problem of the origin of the items:

- The authors point out that they largely chose items from other questionnaires to measure emotion regulation strategies, e.g. for cognitive reappraisal the item chosen was: ‘I tried to think about the situation differently.’ from the ‘reappraisal subscale’ of the Emotion Regulation Questionnaire (ERQ) by Gross, John (2003). The item ‘I made sure not to express what I was feeling’ according to the authors was selected from the same questionnaire (ERQ) to determine the Expressive suppression strategy. The ERQ questionnaire is very well known. Although the authors' items are somewhat related in content to the items in the questionnaire, I did not find items in the ERQ with the same content as presented by the authors (see Gross, John 2003, Table 2 - 1-reappraisal factor, 2-suppression factor).

- The authors indicate that ‘Acceptance was measured with an item derived from acceptance and mindfulness measures (Baer et al., 2004), “I tried to notice my thoughts and feelings without thinking of them as good or bad”. In the questionnaire in the source given, I also did not find an item with such content as indicated by the authors (see Baer et al., 2004 - Table 1 - Kentucky Inventory of Mindfulness Skills). There is instead a item such as: ‘I make judgments about whether my thoughts are good or bad’.

- The authors also indicate that ‘Rumination was measured with an item derived from rumination measures (Treynor et al., 2003), “I kept thinking about and dwelling on my anxiety.”. I did not find such an item in the scale provided in the literature either (see Treynor et al., 2003 - Table 1 - Ruminative Responses Scale).

It is therefore difficult to know where the items presented come from, and if they are related to the content of items from other questionnaires, what modifications they have undergone.

- methodological problem:

Authors used single items with unknown validity and reliability to measure emotion regulation strategies. Even if we assume that a single item was selected from a validated and standardised questionnaire, the validity and reliability of a single item is unknown. Short versions of questionnaires are often used, but these require full re-validation (regardless of the original version). On the other hand, shortening a subscale of a questionnaire to a single item, which in addition has been chosen arbitrarily, is an oversimplification. Unfortunately, a scale consisting of a single item is often too short to be reliable, and the use of a single item is particularly discouraged when measuring such broad constructs as personality or emotional processes. The authors further pointed out, for example, that a specific question was used to measure emotional intensity. When it comes to emotional process, it is very difficult to even assess the reliability of such a single-question scale (test-retest), as this phenomenon is very dynamic. To measure emotion regulation strategies, I would suggest using a short, but recognised, standardised and validated questionnaire or constructing and validating your own questionnaire where the subscale consists of more than one item.

- The authors further write: ‘These strategies include cognitive appraisal, acceptance, problem solving, emotional suppression, rumination, and avoidance. The first three are categorised as engagement strategies, while the latter three are categorised as disengagement strategies'. - The authors selected some items and created new subscales from them as they saw fit. The authors then also analysed the results in the proposed subscales. This approach is also methodologically questionable. Combining selected questions into new subscales or questionnaires would have required theoretical analysis and psychometric re-validation, which unfortunately the authors did not do. In order to create new subscales from the questions used, it would have been necessary to first provide a theoretical perspective on the rationale for their creation, and then perform a psychometric analysis to verify the validity of the approach used.

- In addition, some questions, unfortunately, need to be more explicit, e.g., the item on intensity of emotion would require clarification of what emotion is meant, e.g., positive, negative, etc.

- The authors further indicate” The number of participants required to compute the multilevel linear model with four predictor variables and two-tailed tests using Gpower3.1 was over 42, which provides sufficient power (i.e., greater than .80) to detect effect sizes ≥ .20.” Unfortunately, there are many more than 4 predictors in the model presented. In the final model presented in Table 3, there are as many as 8 expressions for fixed effects, and even before interaction effects, single predictors (without interaction) should still be included, resulting in a much larger number of predictors. Besides, the sample size should be determined for a specific effect size, among other things. It is not known what the name of the effect proposed in the paper is, and hence, among other things, it is difficult to assess its strength so as to determine the sample size. On another note, it is not known how the sample size was determined for such advanced models taking into account both fixed and random effects using Gpower3.1 software.

Results

- The authors indicate that: “Using ecological momentary assessment, we followed 57 athletes during competitions for a period of 6 days before the competition and 2 days during the competition” and then write that a multilevel linear model was used to analyze the longitudinal data. The main purpose of longitudinal research and analysis is to analyze changes over time in relation to various variables. Unfortunately, the authors did not model the time factor, and the data was analyzed as if it had been collected at one time. So I suggest including time factor as a fundamental element in the analysis of longitudinal studies. A good source for longitudinal analysis in multilevel modeling is a widely available textbook: Heck, R. H., Thomas, S. L., & Tabata, L. N. (2013). Multilevel and longitudinal modeling with IBM SPSS. Routledge.

- In addition, the models presented are problematic from a statistical point of view. The authors only presented interaction effects in fixed effects in final model 3, and they should estimate and present all terms (including also effects for single variables without interaction).

6. PLOS authors have the option to publish the peer review history of their article (what does this mean? ). If published, this will include your full peer review and any attached files.

**Do you want your identity to be public for this peer review?** For information about this choice, including consent withdrawal, please see our Privacy Policy .

Reviewer #1: No

Reviewer #2: No

Reviewer #3: No

---

## [Author Response · Author response to Decision Letter 1]

28 Oct 2024

Response Editor,

Thank you for your review of our manuscript. We have carefully considered the feedback from you and the reviewers, and have made comprehensive revisions and adjustments to the entire text. We have addressed the four main issues with significant modifications and have uploaded the manuscript with track changes as requested. We have responded to each reviewer's questions point by point.

Additionally, we have conducted further literature reviews, integrating the opinions of the reviewers with previous research to address each question. The revisions made to these issues have enhanced our understanding of the article, and we are grateful for your and the reviewers' valuable input.

Reviewer #1

Response：We are very grateful for your comments regarding our manuscript “Effect of Emotional Regulation on Performance of Shooters during Competition: an Ecological Momentary Assessment”  All your suggestions are very important to us, both for composing the manuscript and our further research. We have studied comments carefully and have made corrections which we hope meet with approval.  Based on your advice, we amended the relevantsection in the manuscript. All your questions are answered below.

1. Consider revising the Abstract to more accurately reflect the key points and results of your research.

Response: Thank you for your insightful advisement. We have revised the statements regarding the key points and results in the abstract.

2. You refer to the flexibility in the usage of emotion regulation strategies. However, it is unclear which results of your study indicate high or low flexibility. Is it the sum score of the usage of all strategies? Conclusions regarding overall flexibility would require calculating a variable that numerically captures this construct. I am not sure if your data can provide information on emotional flexibility. Additionally, interpreting this in absolute terms would be difficult, as you would require a non-athlete control group to determine if your sample is indeed less flexible in their usage of emotional control strategies. Therefore, I suggest that you only interpret differences between your subgroups (gender and skill level). If you cannot operationalize the flexibility in emotional regulation, please avoid jumping to conclusions (e.g., “Athletes had low flexibility in emotion regulation …” [Abstract], “Low-level athletes have lower emotional regulation flexibility compared to high-level athletes, and they demonstrate inflexibility in both situational demands and specific” [4. Limitations], etc.).

Response: We are very grateful for your valuable suggestions. The calculation of emotion regulation flexibility is not about the total score of all strategies used. Emotion regulation flexibility is primarily reflected in the process by which individuals select appropriate matching strategies based on contextual demands (Naragon-Gainey et al., 2017; Martins et al., 2018; Sheppes & Meiran, 2008; Ortner & Pennekamp, 2020). Therefore, we have readjusted the calculation indicators of emotion regulation flexibility to make them more explicit. Previously, the calculation method separately computed and incorporated contextual demands and strategies into the model. Now, integrating the two into one indicator to reflect emotion regulation flexibility is more reasonable. We have also revised the limitations section regarding the comparison of emotion regulation flexibility among athletes of different skill levels.

3. You use some terminology from shooting sports that laypeople (such as myself) might find difficult to understand. Specifically, please elaborate on the outcomes of your models:

- What is the “proportion of good ten rings” [2.4 Analysis] exactly?

- What is meant by “speed of grip force” [Section 1.1] and how is it related to shooting performance?

- What exactly is meant by “slow the production of force” [Section 3.2] and again, how does this relate to performance?

Additionally, in Section 1.3 you refer to “rapidly changing competitive sports situations.” What do you mean by that? How is this related to shooting (i.e., what are the changing situations in a shooting competition)?

Response：Thank you very much for your invaluable suggestions, esteemed expert. I have incorporated explanations regarding the evaluation indicators of shooting performance into the corresponding sections of the text. Specifically, in Section 2.4, I have added introductions to the concepts and measurement methods of shooting ring value and the "proportion of good ten ring," as well as the data acquisition approaches. Additionally, based on a review of the original literature, I have further elaborated on "speed of grip force" in Section 1.1 and "slow the production of force" in Section 3.2. In Section 1.3, I have also included further clarifications of related concepts to rectify any previously unclear explanations. Your guidance has significantly enhanced the quality and clarity of the text.

4. Regarding your power analysis: If I am not mistaken, gPower is not able to perform power analyses for multilevel linear models. It appears that you computed a power analysis for a linear multiple regression with 4 predictors, resulting in the reported detectable effect sizes of f²=0.20 (I assume that you refer to the effect size parameter f², please make so explicitly). However, a power analysis based on a (non-hierarchical) linear regression is not suitable for a multilevel analysis.

Please conduct a suitable power analysis (for example, using a simulation-based approach) or use another method of justification for your sample size. You may find the following resource helpful:

https://online.ucpress.edu/collabra/article/8/1/33267/120491/Sample-Size-Justification
https://shiny.ieis.tue.nl/sample_size_justification/

Response: Thank you very much for the resources and guidance you provided. We have carefully reviewed the relevant papers and instructional methods to correct our previous incorrect method for sample size calculation. Regarding the power analysis for MLM (multilevel modeling), we utilized the powerSim() function to simulate sample size calculation using the Monte Carlo method (Bolger & Laurenceau, 2013).

5. Please report how many surveys were completed and the extent of dropout you experienced. Additionally, specify if you had any exclusion criteria for the data, such as excluding participants with a survey completion rate below a certain threshold.

Response: Thank you for the reviewer's suggestions. We have incorporated the criteria for data exclusion in Section 2.1.

6. Please provide all survey questions asked in the 2.2 Measures section. The questions for the triggering event and the emotional intensity are missing.

Response: Thank you for the rigorous review, expert. We have supplemented the section on Contextual Demands with items related to triggering events and emotional intensity in the manuscript.

7. Please report the nationalities of the athletes.

Response: Additional clarification regarding the athletes' nationalities has been included in Section 2.1 of the manuscript.

8. The main effect models are interesting, but what about Level 1 interactions? At least two-way interactions might yield interesting results, especially in light of the literature, which has shown that engagement strategies are most effective in low-intensity emotional contexts, while disengagement strategies are more effective in high-intensity emotional contexts. You make several remarks regarding such an interaction:

- “Strategies that require more participation and effort, such as cognitive reappraisal, may be most effective in regulating low-intensity emotions, whereas strategies that involve disengagement from stimuli, such as distraction, may be most effective in regulating high-intensity emotions” [Section 1.3]-“In addition, engagement strategies are more effective at regulating low-intensity emotions, but require a greater allocation of cognitive resources” [Section 3.2]

Exploring these interactions in your models could provide a deeper understanding of how emotion regulation strategies function under varying emotional intensities. I would thus suggest to include a fourth model which encapsulates all two-way interactions of level 1 variables. I would thus suggest including a fourth model that encapsulates all two-way interactions of Level 1 variables. This additional analysis could provide a deeper understanding of how different emotion regulation strategies function under varying emotional intensities and contexts.

Response: Thank you very much for your detailed guidance. The interaction between situational demands and regulatory strategies has a significant impact on athletic performance, which is the focus of our research. It has been effectively concluded in previous studies that the engagement strategy is more effective under high situational demands, while the disengagement strategy is more effective under low situational demands. However, the impact of these interactive effects on athletic performance is the issue that this study is more concerned with. We have conducted a comprehensive discussion and repeated research, and made adjustments in the model part of the data analysis. Situational needs and emotional regulation strategies are considered as first-level variables, and different skill levels are considered as second-level variables. We analyze the impact of this interaction on athletic performance, while also testing whether this impact is consistent across groups of different athletic levels.

9. To strengthen the analysis, I suggest performing a model comparison analysis. Since Model 1 is not nested within Model 2, using an Information Criterion like the Akaike Information Criterion (AIC) and/or the Bayesian Information Criterion (BIC) would be appropriate. These criteria allow for the comparison of non-nested models by evaluating the trade-off between model fit and complexity.

Response: Thank you for your valuable suggestions, which have greatly supported my systematic understanding of the differences between report models. In future studies of this kind, I will complete my thesis report according to your advice. As we have currently overhauled the data analysis model, there is no comparative information between multiple models presented at the moment.

10. At the moment, you only report the random intercept. Could you also report the random effects of gender and (skill-)level?

Response: After reviewing the literature, we have changed our previous data analysis method. Considering that the article focuses on the relationship between situational demands and strategies in emotional regulation (two important processes of emotional regulation), we have integrated indicators of emotional regulation flexibility to more effectively reflect the dynamic relationship between emotional regulation and athletic performance. We have altered the analysis model used before, adding reports of random effects of skill level in the new analysis. Since previous studies have shown that gender only differs in emotional regulation strategies, considering that the research focus is on the flexibility of emotional regulation and shooting performance, which is more likely to be affected by different levels of athletic skill. Therefore, gender is included in the analysis as a control variable.

11. I noticed some inconsistencies in your descriptives, specifically in Table 1: The average ring value is approximately 10, or exactly 10.00 for male athletes. Isn’t this the highest possible value achievable, meaning that they shot the center of the target almost every time (I apologize, I am a layperson regarding shooting)? Is this correct? Could there be a ceiling effect in accuracy? Additionally, how does this align with the ten-ring ratio being approximately 35%? Shouldn’t this be close to 100% if the average ring value was 10? Please clarify these points and explain the potential discrepancy between the average ring value and the ten-ring ratio.

Response: Thank you for your valuable feedback. The scoring rings on the shooting target range from the outer 6-ring to the central 10-ring, with each whole number ring being divided into 10 parts (from 6.0 to 6.9, and so on, with the central 10-ring ranging from 10.0 to 10.9). The precise ring value recorded for the shooter includes a decimal point. A "good ten" usually refers to a score above 10.0. We have added this explanation in the section of the paper that measures shooting performance.

12. “National-level athletes have higher performance levels and significant differences in emotional experience and use of regulatory strategies, which may account for the significant differences in shooting performance.” [Section 3.1] This conclusion is likely confounded with athlete experience and habituation. National-level athletes have probably competed in many competitions and might have habituated to the emotional distress experienced during competition. Their experience might also be a reason for the better performance, raising the question of whether emotion regulation strategies are indeed the primary reason behind their greater success. Please discuss these results more carefully.

Response: Thank you for this valuable suggestion. We have revised some of the content in the discussion section.

13. The discussion of gender differences in Section 3.1 is somewhat brief. Please elaborate further on the differing results for females compared to males. What are the implications of these findings, and why do these differences exist?

Response: Thank you for your insightful question. Regarding gender differences, past research has primarily shown that they manifest in emotional regulation, such as women experiencing emotions more intensely and having a larger repertoire of emotional regulation strategies. However, in terms of overall emotional regulation capabilities, there is no significant gender difference. In our previous analysis, we did include gender as a variable in the model. But in our current revision of the paper, considering that the focus of the study is on the relationship between emotional regulation flexibility and athletic performance, we are treating gender as a control variable. Instead, we are exploring the impact of emotional regulation flexibility on athletic performance across athletes of varying skill levels to see if there are differences. Comparing this influence between genders seems unreasonable since there is no significant gender difference in emotional regulation capabilities and shooting performance. We have added this explanation in the section of the paper that discusses the measurement of shooting performance.

14. “However, the intensity and controllability of emotions are averaged, making it difficult to examine the variability and degree of adaptation of emotional regulation strategies under different situational demands.” [Section 4. Limitations] I do not understand this limitation. You used a hierarchical model for your analysis, so you did not average over intensity and controllability of emotions, correct?

Response: The original statement may lead to ambiguous interpretations. In terms of data analysis, there is no averaging of the intensity and controllability of emotions. Athletes assess emotional events and intensity at a fixed time each day, which can lead to a subjective perception that has been averaged, rather than reflecting the real-time emotional experiences and strategic choices during competitions. To clarify this point, we have revised the statement "However, athletes' retrospective assessment of emotional induction, intensity, and controllability is an evaluation of the overall situation over a short period, not the emotional intensity and controllability that occur in real-time during training and competitions. This also makes it difficult to examine the dynamic changes in athletes' emotional regulation strategies and situational demands over this period. This is an inherent limitation of survey research, which is the inability to record more detailed data more densely."

15. Please discuss the limitations in more detail. For instance, you relied solely on retrospective self-reported data. This requires athletes to a) remember the

---

## [Decision Letter · Decision Letter 1]

28 Nov 2024

PONE-D-24-25633R1Effect of Emotional Regulation on Performance of Shooters during Competition: an Ecological Momentary  Assessment StudyPLOS ONE

Dear Dr. Hua,

Thank you for submitting your manuscript to PLOS ONE. After careful consideration, we feel that it has merit but does not fully meet PLOS ONE’s publication criteria as it currently stands. Therefore, we invite you to submit a revised version of the manuscript that addresses the points raised during the review process. Editor comments. The same reviewers have provided additional feedback on your manuscript. R1 and R2 are satisfied with the revisions and recommend acceptance, whereas R3 remains dissatisfied, raising significant concerns that they feel have not been adequately addressed. Consequently, R3 recommends rejection and does not support the manuscript in its current form. Given these conflicting recommendations, I must make a difficult decision. While I cannot accept the manuscript at this stage due to R3’s unresolved concerns, I believe that a further round of revision could potentially address or mitigate these issues. I am therefore inviting you to submit another revision of your manuscript. If you feel able to address the raised issues, please provide a detailed point-by-point response to all remaining comments from R3. Wherever feasible, incorporate additional analyses or clarifications. For issues that cannot be directly addressed, explicitly discuss and acknowledge these limitations. I hope this further opportunity enables you to resolve the outstanding issues and advance your manuscript, but see my detailed comments below.

We look forward to receiving your revised manuscript.

Kind regards,

Michael B. Steinborn, PhD

Section Editor

PLOS ONE

Additional Editor Comments:

In the following, I will provide (co-)comments on the raised issues and specific aspects of your work, offering feedback from the perspective of an interested reader who is not directly involved in your specific field of study. My comments are intended to support you in revising the manuscript. Given that some issues appear fundamentally unresolvable without new data collection or substantial reworking of the analysis, I aim to provide guidance on how these might be addressed in a way that strengthens the paper. The feedback focuses primarily on objectively identifiable issues rather than engaging in internal theoretical or methodological debates within the field. My intention is to assist in improving the manuscript, not to criticise your work.

(-1-) Origin of items.

R3 has raised concerns about discrepancies between the questionnaire items used in this study and the original validated scales. While modifications do not inherently invalidate an instrument, their appropriateness depends on item content and face validity. I recommend providing justification to ensure theoretical and psychometric soundness. To address this, it may be helpful to detail the origin of the chosen measures, the rationale for modifications, and any supporting theoretical or empirical evidence. If items were adapted to fit the context of this study (e.g., because brief measures were needed to avoid extending testing time unduly), you might explain how these changes align with the construct being measured. I further suggest acknowledging deviations from the original scales and discussing why alternative validated instruments were not used. If evidence, such as pilot testing, supports the validity of these items, I suggest referring to these literature sources, where such evidence is unavailable, however, acknowledging limitations and outlining strategies to mitigate them would enhance transparency. A clear summary of the modifications and their rationale is likely to address R3's concerns and improve methodological rigour.

(-2-) Use of Single-Item Measures.

Single-item measures, though often criticised, are not inherently invalid solely due to the absence of aggregation. From a measurement theory perspective, statistical aggregation typically enhances reliability. However, the validity and reliability of single-item measures depend heavily on their content. For instance, a single-item measure for a straightforward construct like body height in centimetres is likely reliable because individuals vary sufficiently to produce distinguishable values and because it is likely that they will report consistent results in test-retest scenarios (because persons typically know what their height is). This means, whether the choice of items can be justified likely depends on content validity, that is, of how well the items chosen represent the concept being measured. I am not an expert on the specific case at hand, but as a rule of thumb, some constructs are clear and self-evident, making single-item measures potentially justifiable. However, I agree with R3 that complex umbrella concepts, such as emotion regulation strategies, are relatively complicated to represent using single item scores. This is because these constructs encompass numerous dimensions and are not easily captured in a few items, and indeed, even multi-item scales may struggle to fully represent such constructs, as their definitions are context-dependent, thus can vary and encompass a wide range of aspects. Since (as it seems to me) collecting new data is likely not possible or feasible, I recommend more deeply analysing the content validity of the existing items (what do they mean, actually) to evaluate their appropriateness and effectiveness. Such an analysis could address R3’s concerns, at least a bit.

To best support you in the review process, I recommend consulting the abovementioned works, as their methodologies and discussions offer valuable insights relevant to R3’s concerns. While not directly addressing performance anxiety in shooters, they provide a range of specific guidance that is broadly relevant and potentially helpful across topics. This relevance is why I suggest referring to these studies. For instance, Deng et al. (2022) proposed a systematic seven-step plan for evaluating measurement accuracy, covering molecular to molar aspects. I believe this comprehensive approach could serve as a narrative model for elaborating on psychometric aspects of your measures, particularly pertaining to creating new subscales and the use of single-item measures (doi: 10.1016/j.actpsy.2022.103789; doi:10.3389/fpsyg.2022.946626). Implementing this framework may help address R3’s concerns about the psychometric soundness of your instruments. Steghaus and Poth (2024) offer an in-depth analysis of the significance of context and content validity in selecting items for state and trait measurements, directly addressing issues raised about item origin and single-item measures. Aligning your item selection and measurement strategies with their principles could strengthen the theoretical justification for your methods. Similarly, Kärtner et al. (2021) also theorises on the importance of context in measurement and the need for thorough validation when adapting instruments. While not directly related to emotion regulation in shooting competitions, their methodological rigour can inform your approach to justifying item modifications and creating new subscales. Drawing on their validation processes may help reinforce your methodological framework. Incorporating these studies’ methodologies and discussions into your manuscript can provide a solid foundation to address R3’s concerns, demonstrating rigorous measurement practices and clear justifications for your choices, thereby enhancing the validity and credibility of your findings.

Deng, Y. et al. (2022). The effect of mind wandering on cognitive flexibility is mediated by boredom. Acta Psychologica, 231, 103789. doi:10.1016/j.actpsy.2022.103789

Steghaus, S., & Poth, C. H. (2024). Feeling tired versus feeling relaxed: Two faces of low physiological arousal. PLoS One, 19(9), e0310034. doi:10.1371/journal.pone.0310034

Kärtner, L. et al. (2021). Positive expectations predict improved mental-health outcomes linked to psychedelic microdosing. Scientific Reports(1941). doi:10.1038/s41598-021-81446-7

(-3-) Creation of New Subscales

The reviewer raises concerns about the creation of new subscales by combining selected items, emphasising that this approach requires a clear theoretical rationale and psychometric re-validation, which they feel has not been adequately addressed. Their point suggests that items may not reliably measure the intended construct when removed from their original context, as the contextual embedding could influence their meaning. While I cannot be certain of the extent to which this applies in the current study, I recommend that you carefully consider and provide evidence or reasoning to demonstrate that the selected items appropriately measure the intended constructs. For example, a detailed theoretical justification for the combination of items could strengthen the argument for the new subscales, even if re-validation is not feasible. Additionally, clarifying potentially vague items (e.g., emotional intensity) and discussing the limitations of single-item measures explicitly within the manuscript could improve transparency. Providing references to support the decision to use single-item measures in cases where multi-item scales are not practical may also help address this concern. If possible, supplementary analyses or validations could further enhance the reliability of the measures used.

(-4-) Statistical Issues.

There is one issue that R3 raised, which concerns the inclusion of the time factor as a key variable in the multilevel model, suggesting that its omission undermines the longitudinal nature of the data. R3 also recommend presenting all terms in the model, including main effects and interaction effects, rather than focusing solely on interactions. To be honest, I cannot fully assess this issue myself, given that I am not the ultimate expert here, and also as R1 and R2 found the modelling satisfactory and even commended it. However, R3 has provided arguments that seem also plausible to me, and given his expertise in this area, I am inclined to trust his judgment. Therefore, I would tentatively suggest that you carefully consider how the time factor might be incorporated into the model, if feasible. If it is not possible or does not make sense (in your opinion) to include this variable, then providing a clear justification for its exclusion would be good. In any case, acknowledging any limitations and explicitly discussing why certain recommendations cannot be implemented could also help to address R3’s concerns effectively.

Reviewers' comments:

Reviewer's Responses to Questions

**Comments to the Author**

1. If the authors have adequately addressed your comments raised in a previous round of review and you feel that this manuscript is now acceptable for publication, you may indicate that here to bypass the “Comments to the Author” section, enter your conflict of interest statement in the “Confidential to Editor” section, and submit your "Accept" recommendation.

Reviewer #1: All comments have been addressed

Reviewer #2: All comments have been addressed

Reviewer #3: (No Response)

2. Is the manuscript technically sound, and do the data support the conclusions?

Reviewer #1: Yes

Reviewer #2: Yes

Reviewer #3: No

3. Has the statistical analysis been performed appropriately and rigorously? 

Reviewer #1: Yes

Reviewer #2: Yes

Reviewer #3: No

4. Have the authors made all data underlying the findings in their manuscript fully available?

Reviewer #1: No

Reviewer #2: Yes

Reviewer #3: No

5. Is the manuscript presented in an intelligible fashion and written in standard English?

Reviewer #1: (No Response)

Reviewer #2: Yes

Reviewer #3: Yes

6. Review Comments to the Author

Reviewer #1: Thank you very much for your thorough response to my comments. I believe the manuscript has improved considerably and is now ready for acceptance. However, I was unable to access the data via the provided URL. Please ensure that the data is accessible.

Reviewer #2: The manuscript demonstrates a high level of scholarly contribution and is well-prepared for publication. The authors have meticulously addressed all necessary concerns raised, providing comprehensive explanations and ensuring that the content meets the expected standards of rigor and clarity.

Reviewer #3: Unfortunately, the authors did not sufficiently address my main comments. No validity and reliability measures were presented for the questions used to assess emotion regulation strategies. Nor was a psychometric analysis presented for the distinction between engagement strategies and disengagement strategies.

7. PLOS authors have the option to publish the peer review history of their article (what does this mean? ). If published, this will include your full peer review and any attached files.

**Do you want your identity to be public for this peer review?** For information about this choice, including consent withdrawal, please see our Privacy Policy .

Reviewer #1: **Yes: ** Julian Gutzeit

Reviewer #2: No

Reviewer #3: No

---

## [Author Response · Author response to Decision Letter 2]

7 Jan 2025

Thank you very much for the valuable suggestions you have offered regarding this article. I greatly appreciate the detailed and constructive feedback you have provided to improve the manuscript, and I fully agree with your recommendations. Below, I will explain the revisions made in response to the following four aspects one by one.

(-1-) Origin of items.

R3 has raised concerns about discrepancies between the questionnaire items used in this study and the original validated scales. While modifications do not inherently invalidate an instrument, their appropriateness depends on item content and face validity. I recommend providing justification to ensure theoretical and psychometric soundness. To address this, it may be helpful to detail the origin of the chosen measures, the rationale for modifications, and any supporting theoretical or empirical evidence. If items were adapted to fit the context of this study (e.g., because brief measures were needed to avoid extending testing time unduly), you might explain how these changes align with the construct being measured. I further suggest acknowledging deviations from the original scales and discussing why alternative validated instruments were not used. If evidence, such as pilot testing, supports the validity of these items, I suggest referring to these literature sources, where such evidence is unavailable, however, acknowledging limitations and outlining strategies to mitigate them would enhance transparency. A clear summary of the modifications and their rationale is likely to address R3's concerns and improve methodological rigour.

Response: We are deeply appreciative of the valuable feedback provided by the editor and reviewers. In the literature review section, we have meticulously reorganized the classification of emotion regulation strategies and the research methods pertaining to emotional regulation flexibility. We have further clarified the theoretical and empirical basis for our measurement approach. We have also provided a detailed explanation of the relationship and changes between the current measurement items and the original scales. In the discussion and limitations sections, we have added an explanation regarding the insufficiencies of the measurement items and tools selected for this study, and we have suggested areas for improvement in future research.

(-2-) Use of Single-Item Measures.

Single-item measures, though often criticised, are not inherently invalid solely due to the absence of aggregation. From a measurement theory perspective, statistical aggregation typically enhances reliability. However, the validity and reliability of single-item measures depend heavily on their content. For instance, a single-item measure for a straightforward construct like body height in centimetres is likely reliable because individuals vary sufficiently to produce distinguishable values and because it is likely that they will report consistent results in test-retest scenarios (because persons typically know what their height is). This means, whether the choice of items can be justified likely depends on content validity, that is, of how well the items chosen represent the concept being measured. I am not an expert on the specific case at hand, but as a rule of thumb, some constructs are clear and self-evident, making single-item measures potentially justifiable. However, I agree with R3 that complex umbrella concepts, such as emotion regulation strategies, are relatively complicated to represent using single item scores. This is because these constructs encompass numerous dimensions and are not easily captured in a few items, and indeed, even multi-item scales may struggle to fully represent such constructs, as their definitions are context-dependent, thus can vary and encompass a wide range of aspects. Since (as it seems to me) collecting new data is likely not possible or feasible, I recommend more deeply analysing the content validity of the existing items (what do they mean, actually) to evaluate their appropriateness and effectiveness. Such an analysis could address R3’s concerns, at least a bit.

To best support you in the review process, I recommend consulting the abovementioned works, as their methodologies and discussions offer valuable insights relevant to R3’s concerns. While not directly addressing performance anxiety in shooters, they provide a range of specific guidance that is broadly relevant and potentially helpful across topics. This relevance is why I suggest referring to these studies. For instance, Deng et al. (2022) proposed a systematic seven-step plan for evaluating measurement accuracy, covering molecular to molar aspects. I believe this comprehensive approach could serve as a narrative model for elaborating on psychometric aspects of your measures, particularly pertaining to creating new subscales and the use of single-item measures (doi: 10.1016/j.actpsy.2022.103789; doi:10.3389/fpsyg.2022.946626). Implementing this framework may help address R3’s concerns about the psychometric soundness of your instruments. Steghaus and Poth (2024) offer an in-depth analysis of the significance of context and content validity in selecting items for state and trait measurements, directly addressing issues raised about item origin and single-item measures. Aligning your item selection and measurement strategies with their principles could strengthen the theoretical justification for your methods. Similarly, Kärtner et al. (2021) also theorises on the importance of context in measurement and the need for thorough validation when adapting instruments. While not directly related to emotion regulation in shooting competitions, their methodological rigour can inform your approach to justifying item modifications and creating new subscales. Drawing on their validation processes may help reinforce your methodological framework. Incorporating these studies’ methodologies and discussions into your manuscript can provide a solid foundation to address R3’s concerns, demonstrating rigorous measurement practices and clear justifications for your choices, thereby enhancing the validity and credibility of your findings.

Response: We are grateful to the editor and reviewers for their valuable suggestions to improve this manuscript. Special thanks to the editor for the constructive feedback provided for further enhancement. We have revisited the literature and organized the measurement of the psychological variable of emotion regulation, seeking new measurement rationales to substantiate the validity of the measurement tools employed in this study.

Emotion regulation possesses distinct characteristics, differing from previous examinations of the relationship between emotional regulation tendencies (strategy use preferences) and environmental adaptation. This study focuses on investigating whether emotional regulation flexibility promotes athletic performance. Conventional emotion regulation measurement tools can only measure individuals' preferences for emotion regulation strategies and cannot reflect the potential for individuals to use a variety of different strategies based on situational changes. Emotional regulation tendencies and emotional regulation flexibility are two different features of emotion regulation, with significant conceptual differences. Original measurement tools tend to measure individuals' preferences for using strategies in relatively stable situations. Although the core characteristics of strategies themselves are stable (e.g., reappraisal strategies involve changing thoughts), the regulatory context varies greatly and is dynamically changing. Therefore, examining individuals' strategy use in response to situational changes is an investigation of short-term psychological changes. When assessing short-term psychological changes such as relaxation, the corresponding measurement tools should focus on the variability of context and content (Stegaus & Poth, 2024; Kärtner et al., 2021). According to previous research, tools and items measuring short-term psychological changes should possess sensitivity (to capture state changes), contextuality (to consider the variability of measurement background or context), and individual variability (to adapt measurement items to different individuals).

Based on the capability model of emotion regulation, the core characteristics of the six regulatory strategies are relatively stable. Cognitive reappraisal is fundamentally characterized by "changing thoughts," expressive suppression is primarily manifested as "inhibiting emotional experiences" (Gross & John, 2003), and acceptance is merely noticing without judgment (Baer et al., 2004). Problem solving refers to finding specific methods to deal with current emotions (D’Zurilla & Nezu, 1990). Rumination is characterized by repetitive or immersive experiencing or thinking about certain emotions (Treynor et al., 2003). Avoidance encompasses multiple strategies to escape, control, or suppress unwanted thoughts, emotions, and sensations (Kashdan et al., 2013).

(-3-) Creation of New Subscales

The reviewer raises concerns about the creation of new subscales by combining selected items, emphasising that this approach requires a clear theoretical rationale and psychometric re-validation, which they feel has not been adequately addressed. Their point suggests that items may not reliably measure the intended construct when removed from their original context, as the contextual embedding could influence their meaning. While I cannot be certain of the extent to which this applies in the current study, I recommend that you carefully consider and provide evidence or reasoning to demonstrate that the selected items appropriately measure the intended constructs. For example, a detailed theoretical justification for the combination of items could strengthen the argument for the new subscales, even if re-validation is not feasible. Additionally, clarifying potentially vague items (e.g., emotional intensity) and discussing the limitations of single-item measures explicitly within the manuscript could improve transparency. Providing references to support the decision to use single-item measures in cases where multi-item scales are not practical may also help address this concern. If possible, supplementary analyses or validations could further enhance the reliability of the measures used.

Response: In the literature review section, we have added theoretical underpinnings regarding the measurement of the emotion regulation process. Recently, longitudinal tracking of individuals' dynamic changes in emotion regulation has gradually become an important trend in emotion regulation research. This study focuses on examining the characteristic of emotional regulation flexibility. The core of emotional regulation flexibility lies in the flexible selection and use of regulatory strategies in response to situational changes, whereas traditional emotion regulation measurement tools primarily examine habitual use of emotion regulation strategies without considering situational changes. The essence of the concept of emotional regulation flexibility is to flexibly alter emotion regulation strategies according to the demands of emotion regulation (situational changes). The rules governing the use of emotion regulation strategies themselves do not change with the context, such as cognitive reappraisal, which requires individuals to change their thoughts and interpretations of different eliciting situations. Based on this, the examination of emotional regulation flexibility in this study is grounded in the variability of situational changes and the selection of regulatory strategy use rules that have been widely validated in previous research, to explore the dynamic changes in athletes' emotion regulation processes. In the limitations section of the article, we have added corresponding explanations: Examining the variability of emotion regulation strategies with situational changes, using subjective assessment methods to examine elicited emotions (emotional intensity) is not clear and may be biased; future studies should adopt a combination of subjective and objective methods (such as heart rate, electromyography, and other physiological measurements) to enhance the reliability of the research.

(-4-) Statistical Issues.

There is one issue that R3 raised, which concerns the inclusion of the time factor as a key variable in the multilevel model, suggesting that its omission undermines the longitudinal nature of the data. R3 also recommend presenting all terms in the model, including main effects and interaction effects, rather than focusing solely on interactions. To be honest, I cannot fully assess this issue myself, given that I am not the ultimate expert here, and also as R1 and R2 found the modelling satisfactory and even commended it. However, R3 has provided arguments that seem also plausible to me, and given his expertise in this area, I am inclined to trust his judgment. Therefore, I would tentatively suggest that you carefully consider how the time factor might be incorporated into the model, if feasible. If it is not possible or does not make sense (in your opinion) to include this variable, then providing a clear justification for its exclusion would be good. In any case, acknowledging any limitations and explicitly discussing why certain recommendations cannot be implemented could also help to address R3’s concerns effectively.

Response:We are deeply grateful for reviewers’s meticulous examination of our manuscript and the valuable feedback you have provided. We have carefully considered all your suggestions and have made the necessary revisions. In the previous round of revisions, we have reworked our data analysis methods. R version 3.6.1 was employed to conduct Multi-level Linear Model analysis. A two-tiered model was established with the dependent variables being the average shooting score and the proportion of perfect tens. Time-level (Level 1) linear mixed-effects models were performed using the lme4 package. Following your expert advice, we have incorporated the time factor into our model. We have highlighted the changes in the text using different shades of yellow as suggested. In this analysis, the time factor was included at the Day-level in the first tier of the model, focusing on examining the dynamic relationship between athletes' emotional regulation flexibility and athletic performance.

---

## [Editor Report · Decision Letter 2]

23 Jan 2025

Effect of Emotional Regulation on Performance of Shooters during Competition: an Ecological Momentary  Assessment Study

PONE-D-24-25633R2

Dear Dr. Hua,

We’re pleased to inform you that your manuscript has been judged scientifically suitable for publication and will be formally accepted for publication once it meets all outstanding technical requirements.

Final editor comment.

The authors have thoroughly addressed all comments and significantly improved the manuscript, which is now well-elaborated and in excellent shape. While the fundamental critiques raised by R3 remain partially unresolved, addressing them would require new data collection. However, it remains unclear what specific additional data would sufficiently resolve these concerns, leaving the matter open to interpretation. Within these constraints, the authors have made every effort to address the concerns as comprehensively as possible. After careful consideration, I conclude that the manuscript is now ready for publication in its present form.

Kind regards,

Michael B. Steinborn, PhD

Section Editor

PLOS ONE
---

## [Editor Report · Acceptance letter]

PONE-D-24-25633R2

PLOS ONE

Dear Dr. Hua,

I'm pleased to inform you that your manuscript has been deemed suitable for publication in PLOS ONE. Congratulations! Your manuscript is now being handed over to our production team.

Kind regards,

on behalf of

Dr. Michael B. Steinborn

Section Editor

PLOS ONE